

# Concurrent design of control software and configuration of hardware for robot swarms under economic constraints

Muhammad Salman, Antoine Ligot and Mauro Birattari

IRIDIA, Université Libre de Bruxelles, Brussels, Belgium

## ABSTRACT

Designing a robot swarm is challenging due to its self-organized and distributed nature: complex relations exist between the behavior of the individual robots and the collective behavior that results from their interactions. In this paper, we study the concurrent automatic design of control software and the automatic configuration of the hardware of robot swarms. We introduce Waffle, a new instance of the AutoMoDe family of automatic design methods that produces control software in the form of a probabilistic finite state machine, configures the robot hardware, and selects the number of robots in the swarm. We test Waffle under economic constraints on the total monetary budget available and on the battery capacity of each individual robot comprised in the swarm. Experimental results obtained via realistic computer-based simulation on three collective missions indicate that different missions require different hardware and software configuration, and that Waffle is able to produce effective and meaningful solutions under all the experimental conditions considered.

## INTRODUCTION

In this paper, we make a two-fold contribution: (i) we address the concurrent automatic design of control software and the automatic configuration of the hardware; and (ii) we study an automatic design problem that is subject to economic constraints.

In swarm robotics (*Şahin, 2004*), a group of robots coordinate to accomplish missions that a single robot cannot accomplish. In a swarm, robots do not have predefined roles, neither do they possess any global information, nor do they rely on external infrastructures (*Dorigo, Birattari & Brambilla, 2014*). They operate in a decentralized and self-organized manner, relying only on local information gathered through their sensors or communicated by their neighboring peers. A robot swarm is a collective entity and cannot be programmed directly: the designer must program the individual robots so that, together, they perform the desired mission. The design process is laborious due to the complex relation existing between the behavior of the individual robots and the collective behavior that results from their interactions (*Brambilla et al., 2013*). The most common approach to designing a robot swarm is trial-and-error: a time consuming approach in which individual-level behaviors are implemented, tested, and modified until

Corresponding author
Mauro Birattari, mbiro@ulb.ac.be

the desired swarm-level behavior is obtained. Although a number of swarms have been successfully designed with this approach, it heavily depends on the experience of designer, it is error-prone, and its results are not reproducible. To overcome these issues, a number of principled manual design methods have been proposed. However, these methods are limited in scope: a universal swarm design methodology does not exist, yet (*Hamann & Wörn, 2008*; *Lopes et al., 2016*; *Brambilla et al., 2015*). Automatic design is an alternative approach to designing a swarm. In automatic design, the design problem is formulated as an optimization problem that is then solved with an optimization algorithm (*Birattari et al., 2019*). A design problem of a collective mission is expressed as an objective function, a mathematical equation that measures the performance of the robot swarm. An optimization algorithm steers the search for a control software of an individual-robot that maximizes the performance of the swarm, taking into account the constraints such as hardware limitations of the robots or other environmental restrictions, that are encoded in the form of additional (in)equalities. Neuro-evolutionary robotics is the most studied among the existing automatic design approaches to design a swarm (*Trianni, 2008*). This approach uses an evolutionary algorithm to optimize the control software of robots that, in this case, is represented by an artificial neural network (*Nolfi & Floreano, 2000*). Recently, a number of automatic design approaches have been proposed that use different control software structures and optimization algorithms than those typically adopted in evolutionary swarm robotics (*Francesca et al., 2014*).

The concurrent development of control software and hardware is a further step to reduce the human involvement in the design process. A number of concurrent design methods have been proposed for single-robot applications: in addition to designing the control software, they select and configure sensors and actuators and/or the robot's morphology (*Sims, 1994*; *Lipson & Pollack, 2000*). These concurrent design methods have significantly increased the performance and versatility of the designed robots. In swarm robotics, only a few research articles have been published that studied the concurrent automatic design of control software and configuration of the hardware (*Watson & Nitschke, 2015*). Details are provided in the **STATE OF THE ART** section.

In general, designing implies solving trade-offs, that is, balancing multiple, possibly conflicting factors (*Pahl et al., 2007*). In swarm robotics, a characterizing example of a trade-off to be handled is the one between the number of robots comprised in the swarm and the capabilities of each individual robot. The designer must decide whether, for the specific mission at hand, they should use (i) few highly capable robots or (ii) many relatively incapable ones. This trade-off originates from the constraint of a limited monetary budget. Indeed, it is reasonable to assume that highly capable robots are more expensive than relatively incapable ones. Another example of a design trade-off originates if the designer is given the constraint of adopting a battery of a predefined capacity. Under this constraint, the designer might chose to adopt (i) robots with powerful sensors and actuators that can achieve relatively more per unit time, but have a high power consumption and therefore can operate for a relatively short amount of time; or (ii) simpler robots that can achieve relatively less per unit time but have a low power consumption and therefore can operate

for a relatively long amount of time. It is reasonable to expect that the choice might depend on the specific mission at hand.

In this research, we introduce `Waffle`, a new instance of the AutoMoDe family of automatic design methods. All previously published instances of AutoMoDe generate control software for the e-puck platform (*Mondada et al., 2009*) by selecting, combining, and fine-tuning predefined, mission-independent software modules (*Francesca et al., 2014*; *Francesca et al., 2015*; *Kuckling et al., 2018*; *Hasselmann, Robert & Birattari, 2018*). `Waffle` is based on `Chocolate` (*Francesca et al., 2015*). Indeed, regarding the conception of control software, `Waffle` is identical to `Chocolate`: the two methods share the same predefined software modules, they combine these modules into the same control software architecture, and they use the same optimization algorithm—details are given in the **AUTOMODE-Waffle** section. The novelty of `Waffle` is the concurrent hardware configuration of the robot swarm: `Waffle` automatically selects the hardware configuration of the individual robots and the number of robots within the swarm. The goal of this research is to show that it is possible to concurrently design the control software and configure the hardware for robot swarm using the principles of automatic modular design: the idea that control software and, in our particular case the hardware, can be produced by combining pre-existing modules that are mission independent and that are assembled and fine tuned automatically. In this specific study, we consider some hypothetical hardware modules that enable a robot to detect and locate its neighboring peers. These hypothetical modules are based on infrared transceivers and are variants of an existing hardware module for the e-puck platform (*Mondada et al., 2009*) known as the range-&-bearing (*Gutiérrez et al., 2009*). We define the set of these hypothetical modules so that some of them are more-capable and some are less-capable than the existing one in terms of perception range and detection abilities. We assume that the more capable hardware modules are more expensive and consume more power. These hypothetical modules are realistic and possibly implementable. The fact that they are hypothetical (except one) does not impair the significance of our research. Indeed, our goal is not to solve a specific design problem but rather to show that a modular approach to designing by optimization can search the space of possible hardware configurations concurrently with the automatic design of control software. We study `Waffle` under what we shall collectively call *economic constraints*, namely, constraints on the total monetary budget available and on the battery capacity of each individual robot comprised in the swarm. If these constraints were not included, the study would produce trivial results in many cases. Indeed, in many cases, the automatic design process would produce swarms comprising the largest number of robots possible, each equipped with the best performing, most expensive, and most energy-consuming hardware modules. Besides preventing that the study produces trivial results, these constraints have a value on their own. Indeed, in a prospective practical application of automatic design, one will be necessarily faced with economic constraints, which are an essential, unavoidable element of any real-world design problem. To the best of our knowledge, this study is the first one in which automatic design of robot swarms is studied under constraints of economical nature. In this sense, our work contributes to moving a step in the direction of the practical application of automatic design.

The main research question that we address in this paper is the following: can `Waffle` select mission-specific hardware together with an appropriate control software? To do so, we test `Waffle` on three different collective missions: END-TIME-AGGREGATION, ANYTIME-SELECTION, and FORAGING. For each mission, we impose constraints to the design process. Namely, we impose a monetary budget and/or a battery capacity. For each mission, we perform nine different experiments: (i) one experiment in which both monetary budget and battery capacity are unconstrained (*No-Constraint*), (ii) two experiments with different levels of the monetary budget and unconstrained battery capacity (*Monetary-Constraint*), (iii) two experiments with different levels of battery capacity and unconstrained monetary budget (*Power-Constraint*), and (iv) four experiments with different levels of monetary budget and battery capacity (*Monetary-&-Power-Constraint*). For each experiment, we report and discuss (i) a measure of the performance achieved, (ii) the number of robots comprised in the automatically designed swarm, (iii) which hardware modules have been automatically selected, and (iv) which software modules were adopted.

## STATE OF THE ART

In the literature, a number of approaches have been proposed to address the concurrent design of single robots. However, only a few preliminary studies have been published that implement the simultaneous design of hardware and control software for a robot swarm.

In single robot applications, *Sims (1994)* introduced what he called *virtual creatures*: simulated robots whose body and control software are designed simultaneously to perform different tasks, such as walking, jumping, and swimming. The body of these robots is composed of solid cuboid segments connected by different joint types, actuators to simulate muscle force at joints, and various sensors. The body of a robot is represented as a directed-graph of nodes and connections that contain the connectivity information and developmental instructions. The control software of the robot is implemented as an artificial neural network. A genetic algorithm was used to concurrently design the software and the hardware of a robot for a particular task. The development of virtual creatures demonstrated the ability of this approach to design complex systems that would be complicated to design using traditional methods. *Lipson & Pollack (2000)* took this concept to a further level by introducing the automatic manufacturing of the concurrently designed robot. The authors used the rapid prototyping technology to 3D print the robot once its body (variable-length bars, and ball-and-socket joints) and control software (artificial neural network) is automatically designed in the simulation. In recent studies, much work has been conducted using similar approaches that aim to address various design problems, e.g., robots with insect-like hardware topologies and behaviors (*Hornby, Lipson & Pollack, 2003*); visually guided robots (*Macinnes, 2003*); aquatic robots (*Clark et al., 2012*); self-reconfiguring robot (*Nygaard et al., 2018*).

In swarm robotics, only a couple of studies are available that use concurrent design methods to design a robot swarm. *Watson & Nitschke (2015)* studied the impact of the number of sensors and their position on the robot to select the minimal sensor configuration of individual robot for a collective construction task. They achieved that by manually

selecting six different sensors configurations and generating six instances of control software in the form of artificial neural networks using HyperNeat. *Hewland & Nitschke (2015)* used NEAT-M to configure the number and types of sensors simultaneously with the control software for the robots in a swarm for collective construction task. Moreover, they also designed the control software for a robot swarm with fixed hardware configuration. According to the authors, the concurrently designed swarm performed relatively better than the swarm with fixed hardware configuration. *Heinerman, Rango & Eiben (2015)* studied the relationship between individual and social learning in physical robot swarms. The authors used six Thymio II robots in their experiments. The study shows that the on-line social learning in a physical robot swarm is possible, the design process is faster than individual learning, and the performance of the produced control software (artificial neural networks) is higher. Moreover, the design process also configures a suitable sensory layout for individual robots.

Various computational models have been proposed to estimate the impact of the size/density of the robot swarm on its performance (*Lerman & Galstyan, 2002*; *Hamann, 2012*). However, we are not aware of any study in which the automatic selection of the number of robots for a swarm has been attempted. To the best of our knowledge, the implications of imposing economical constraints to the automatic design of a robot swarm have never been studied. We are only aware of a single study that goes into that direction: recently, *Carlone & Pinciroli (2019)* included some practical constraints in the design of a robot swarm. They formulate the co-design of a single race-drone and multi-drone system as a binary optimization problem that allows specifying constraints such as the total design budget.

## AUTOMODE-WAFFLE

As already mentioned above, `Waffle` belongs to AutoMoDe, a family of off-line automatic methods for designing the control software of robot swarms (*Francesca et al., 2014*). In AutoMoDe, control software is generated by automatically assembling predefined modules and by fine-tuning their free parameters. A number of methods have been proposed that belong to AutoMoDe: `Vanilla` (*Francesca et al., 2014*), `Chocolate` (*Francesca et al., 2015*), `Gianduja` (*Hasselmann, Robert & Birattari, 2018*), and `Maple` (*Kuckling et al., 2018*). Each of these methods is characterized by a specific set of predefined modules, a software architecture into which these modules can be combined, and an optimization algorithm that searches the space of the possible ways in which modules can be combined into the given architecture and the space of the free parameters. All the aforementioned methods generate control software for a specific version of the e-puck platform (*Mondada et al., 2009*). Moreover, they all limit themselves to the generation of control software: the hardware configuration of the e-puck robot is fixed and the number of robots comprised in the swarm is given as a requirement of the mission to be performed.

`Waffle` is a further step to increase the flexibility of AutoMoDe and to reduce the human involvement in the design process. Indeed, `Waffle` concurrently develops the control software and configures the hardware of the robot swarm—including the number

**Table 1   Low-level behaviors and conditions used in `Waffle`.**

|  | **Low-level behaviors** |
| --- | --- |
| Exploration | The robot moves straight. If an obstacle is detected, the robot rotates in place for a random amount of time before moving straight again |
| Stop | The robot does not move |
| Phototaxis | The robot moves towards the light, if perceived; otherwise, it moves straight |
| Anti-phototaxis | The robot moves away from the light, if perceived; otherwise, it moves straight |
| Attraction[a] | The robot moves towards peers within its perception range |
| Repulsion[a] | The robot moves away from peers within its perception range |
|  | **Conditions** |
| Black-floor | Black floor is detected |
| Gray-floor | Gray floor is detected |
| White-floor | White floor is detected |
| Neighbor-count[a] | The number of peers in neighborhood is greater than a parameter |
| Inverted-neighbor-count[a] | The number of peers in neighborhood is less than a parameter |
| Fixed-probability | The transition is enabled with a fixed probability |

**Notes.**
[a] Behaviors and conditions that use the range-&-bearing module.

of robots comprised. Regarding the design of control software, `Waffle` is identical to `Chocolate` (*Francesca et al., 2015*): the two methods share the same set of pre-defined software modules; generate control software in the form of probabilistic finite state machines; and use the Iterated F-race optimization algorithm (*López-Ibáñez et al., 2016*) to select, combine, and fine-tune the software modules. The set of software modules is composed of six low-level behaviors and six conditions (*Francesca et al., 2015*). A behavior is an operation or action that a robot can perform, while a condition is a criterion to switch from one behavior to another. Behaviors and conditions have parameters that impact their internal functioning: AutoMoDe fine-tunes these parameters to the specific mission to be performed. Multiple instances of the same behavior might coexist in a probabilistic finite state machine, possibly with different values of the parameters. In `Waffle` (as in `Chocolate`), states and edges of a probabilistic finite state machine are instances of behaviors and conditions, respectively. The design process can include a maximum of four states and each state can have at most four outgoing edges. A brief description of the software modules is given in Table 1 and a typical probabilistic finite state machine is shown in Fig. 1. Regarding the hardware, `Waffle` uses Iterated F-race to define the configuration of the individual e-puck robots and their number within the swarm.

The e-puck is a differential drive robot that is widely used in swarm robotics research (*Mondada et al., 2009*). `Waffle` and all previous instances of AutoMoDe operate with an extended version of the e-puck robot, which adopts: (i) the Overo Gumstix, to run Linux

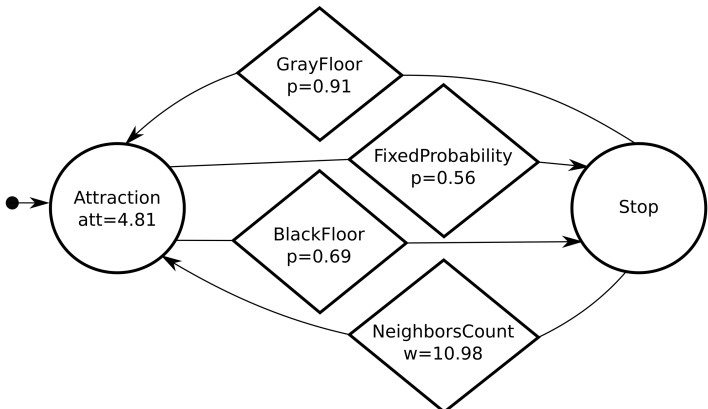

**Figure 1** **A typical probabilistic finite state machine automatically designed by `Waffle`: states and conditions are represented by circles and diamonds, respectively.** Initially the robot moves towards its neighboring peers (attraction state)—the robot follows a direction vector and $att = 4.81$ is the attraction parameter that defines the magnitude of the vector. When it detects the black floor, it stops. The parameter $p$ is the probability of transition from one state to another when the condition is true. We refer the reader to *Francesca et al. (2014)* for further details.

on the robot, (ii) three ground sensors, located under its body, to detect the gray-level color of the floor beneath it, and (iii) a range-&-bearing module (*Gutiérrez et al., 2009*) to detect neighboring peers and have knowledge of their relative position. We simulate the e-puck robots using ARGoS (*Pinciroli et al., 2012*; *Garattoni et al., 2015*), an open source multi-engine simulator for robot swarm. We use ARGoS' 2D dynamic physics engine to simulate the robots and the environment.

Here, we assume that e-puck robots are formally described by reference model RM 1.1 (*Hasselmann et al., 2018*), which defines the input and output variables of corresponding sensors and actuators—see Table 2. These variables can be read/written by the control software at every control step, that is, every 100 ms. The control software detects the obstacles ($prox_i$) and the presence and relative position of a light source ($light_i$) using eight infrared transceivers. It also detects the gray-level color of the floor ($ground_j$) beneath the robot using ground sensors. At every control step, all robots in the swarm broadcast a "heartbeat" signal using their range-&-bearing module. This signal encodes the sender's unique ID. Every robot receives the heartbeat signals of the peers that are in its perception range and has therefore knowledge of their number ($n$), and of their aggregate relative position ($V$) which is defined as:

$$V = \begin{cases} \sum_{m=1}^{n} \left( \frac{1}{1+r_m}, \angle b_m \right), & \text{if robots are perceived;} \\ (1, \angle 0), & \text{otherwise.} \end{cases} \quad (1)$$

Here, $r_m$ and $b_m$ are the range and bearing of the *mth* neighboring peer, respectively. For a detailed description of the vector $V$ and of how it is computed, see *Salman, Ligot & Birattari (2019)*. Eventually, the control software actuates the wheels of the robot by setting the right and left wheel velocity ($v_r$ and $v_l$).

**Table 2  Reference model RM 1.1.**

| Sensor | Input | Value | Description |
|---|---|---|---|
| Proximity | $prox_{i \in \{1,...,8\}}$ | $[0,1]$ | reading of proximity sensor $i$ |
| Light | $light_{i \in \{1,...,8\}}$ | $[0,1]$ | reading of light sensor $i$ |
| Ground | $ground_{j \in \{1,2,3\}}$ | $\{black, gray, white\}$ | reading of ground sensor $j$ |
| Range-&-Bearing | $n$ | $[0,29]$ | number of neighboring robots perceived |
| | $V$ | $([0.5,30],[0,2\pi))$ | their aggregate position |

| Actuator | Output | Value | Description |
|---|---|---|---|
| Motors | $v_{k \in \{l,r\}}$ | $[-0.12, 0.12]$ ms$^{-1}$ | target linear wheel velocity |

As mentioned above, the goal of this research is to concurrently develop the control software and configure the hardware for the robot swarm. Concerning the hardware configuration, Waffle configures the range-&-bearing transceiver modules of e-puck robots. To do so, we simulate six range-&-bearing receivers and two range-&-bearing transmitters as listed in Table 3. These range-&-bearing modules are hypothetical but are variants of one that actually exists (*Gutiérrez et al., 2009*): receiver $R_{rb}^3$ coupled with transmitter $T_{rb}^1$, as defined in Table 3. Each hypothetical range-&-bearing receiver and transmitter has distinct characteristics. A receiver is characterized by an error modeled as white noise in the estimation of the angle of a broadcasting peer (bearing error), and a probability to fail to receive the signal broadcast by a robot in its perception range (loss probability). The bearing error is sampled at every time step from a uniform distribution. The loss probability is a function of the number of neighboring peers—details are given as supplementary material (*Salman, Ligot & Birattari, 2019*). A range-&-bearing transmitter is characterized by a tunable infra-red transmission range ($\mathcal{R}$)—see Table 3. If the design process finds the range-&-bearing necessary for a mission, it can equip all the robots with one of the receiver and of the transmitter configurations listed in Table 3. In configuring the hardware of the robot swarm, the design process must also respect the available monetary budget and/or a battery capacity. Indeed, the range-&-bearing receivers and transmitters are also characterized by price and current rating—see Table 3.

# EXPERIMENTAL SETUP

In this section, we describe the three collective missions, the experiments we perform for each mission, and the protocol we follow to test Waffle.

## Missions

We test Waffle on three missions: ANYTIME-SELECTION, END-TIME-AGGREGATION, and FORAGING. All three missions are to be performed in a dodecagonal arena of 4.91 m$^2$. The arena is divided into different zones according to the requirements of a mission. ANYTIME-SELECTION and END-TIME-AGGREGATION are performed in the same arena—as shown in Fig. 2A. At the beginning of every experimental run, we randomly position the robots everywhere in the arena.

**Table 3  Extended range-&-bearing receiver and transmitter modules.** The bearing error is modeled as white noise in the estimation of the bearing of a broadcasting peer and is sampled from a uniform probability distribution, of which we list here the extremes of the support. The loss probability is a function of the number of neighboring peers, of which we list here the minimum, average, and maximum values.

| Range-&-bearing Receivers $R_{rb}^x$ | Bearing error $\pm$ deg | Loss probability $min - avg - max$ | Price $P_x$ (€) | Current rating $I_x$ (mA) |
|---|---|---|---|---|
| $\emptyset$ | — | — | 0 | 0 |
| $R_{rb}^1$ | 45 | $0.75 - 0.84 - 0.95$ | 500 | 10 |
| $R_{rb}^2$ | 30 | $0.75 - 0.85 - 0.90$ | 600 | 15 |
| $R_{rb}^3$ | 25 | $0.75 - 0.80 - 0.93$ | 700 | 20 |
| $R_{rb}^4$ | 20 | $0.70 - 0.78 - 0.85$ | 800 | 25 |
| $R_{rb}^5$ | 15 | $0.50 - 0.64 - 0.75$ | 900 | 30 |
| $R_{rb}^6$ | 5 | $0.40 - 0.57 - 0.70$ | 1,000 | 35 |

| Range-&-bearing Transmitters $T_{rb}^y$ | Range $\mathcal{R}$ (m) | Price $P_y$ (€) | Current rating $I_{y(R)}$ (mA) |
|---|---|---|---|
| $\emptyset$ | — | 0 | 0 |
| $T_{rb}^1$ | $\{0.6, 0.7, 0.8\}$ | 400 | $\{20, 30, 40\}$ |
| $T_{rb}^2$ | $\{0.9, 1.0\}$ | 600 | $\{50, 60\}$ |

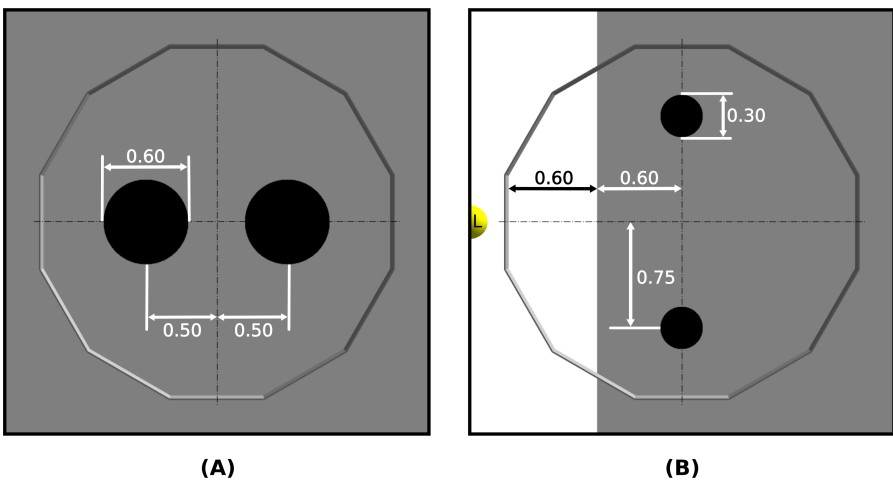

**(A)**    **(B)**

**Figure 2  ARGoS representation of arenas with dimensions and positions of different zones: (A) END-TIME-AGGREGATION and ANYTIME-SELECTION, and (B) FORAGING.** Measurements are expressed in meters. In FORAGING, L represents a light source.

## ANYTIME-SELECTION

The robot swarm must aggregate in one of the two circular black zones. The size of two black zones and their position in the arena are given in Fig. 2A. The performance of the swarm is measured by the following objective function:

$$F_A = \sum_{t=1}^{T} \left| (N_a(t) - N_b(t)) \right|,$$

(2)

where $N_a(t)$ and $N_b(t)$ are the number of robots in zone $a$ and $b$ at any time $t$; $T$ is the total duration of the experiment.

In ANYTIME-SELECTION, the performance is measured at every control step. Due to this reason, the robots are expected to promptly aggregate in one of the black zones and stay there until the end of the experiment.

**END-TIME-AGGREGATION**

The robots must aggregate in one of the two black zones. The dimensions of two black zones and their position in the arena are given in Fig. 2A. The performance of the robot swarm is measured with the following objective function:

$$F_E = \left| N_a(T) - N_b(T) \right|, \tag{3}$$

where $T$ is the duration of an experiment and $N_a(T)$ and $N_b(T)$ are the number of robots in zone $a$ and zone $b$ at time $T$.

Unlike ANYTIME-SELECTION, the performance in END-TIME-AGGREGATION is computed at the end of the experiment. Due to this reason, the robots can take some time to explore the arena and converge in a black zone: the experiment duration is not a significant constraint in this mission. However, the real challenge is to keep the robots assembled in a zone until the end of the experiment.

**FORAGING**

The swarm must collect a maximum number of objects from two sources and drop them in the nest. We abstract the FORAGING experiment by considering that an object is retrieved when an individual robot visits a source, and an object is dropped when a robot visits the nest. The two sources in the arena are represented as two black zones, while the nest is represented as a white zone. A light is also placed behind the nest as a cue for robots. The dimensions and position of the two source zones and nest are given in Fig. 2B. The performance of the robot swam, the number of objects ($N_o$) retrieved by the swarm, is expressed by the following objective function:

$$F_F = N_o. \tag{4}$$

In FORAGING, an individual robot can retrieve a single object at a time. Therefore, the performance of the swarm heavily rely on the number of robots and on the duration of the experiment.

## Experiments

We perform nine different experiments for each mission. In these experiments, we impose a monetary budget and/or a battery capacity constraints to the design process. Depending on the type of constraint, an experiment can be classified as belonging to one of four categories: *No-Constraint*, *Monetary-Constraint*, *Power-Constraint*, and *Monetary-&-Power-Constraint*. The levels of the monetary constraint, levels of battery capacity, and duration of the experiments are listed in Table 4. For each experiment, the design process is free to choose any number of robots between 15 and 30.

**Table 4  Monetary budget levels, battery capacity levels and duration of all nine experiments of four categories.** The duration of an experiment, $T$, from categories *Power-Constraint* and *Monetary-&-Power-Constraint* is not fixed. The experiment terminates, when all robots are out of battery—as defined in Eq. (7).

| Experiment | Category | Monetary budget | Battery capacity | Duration |
|---|---|---|---|---|
| $NC$ | *No-Constraint* | unconstrained | unconstrained | 500 s |
| $M_{80}$ | *Monetary* | 80,000€ | unconstrained | 500 s |
| $M_{60}$ | *constraint* | 60,000€ | unconstrained | 500 s |
| $P_{20}$ | *Power* | unconstrained | 20 mAh | $T$ |
| $P_{15}$ | *constraint* | unconstrained | 15 mAh | $T$ |
| $M_{80}P_{20}$ | *Monetary* | 80,000€ | 20 mAh | $T$ |
| $M_{80}P_{15}$ | *&* | 80,000€ | 15 mAh | $T$ |
| $M_{60}P_{20}$ | *power* | 60,000€ | 20 mAh | $T$ |
| $M_{60}P_{15}$ | *constraint* | 60,000€ | 15 mAh | $T$ |

### No-constraint

This experiment ($NC$) is performed without any constraint: the monetary resources and battery capacity are unconstrained.

### Monetary-constraint

In these experiments, the limited resource is the monetary budget, $M_{limit}$, available to purchase the robots and range-&-bearing modules. The design process only considers the combinations of hardware modules that keep the total cost of the swarm, $P_{swarm}$, equal or below the available monetary budget—i.e, $P_{swarm} \leq M_{limit}$. The total swarm cost, $P_{swarm}$, is computed with the following equation:

$$P_{swarm} = N \times (P_r + P_x + P_y),  \tag{5}$$

here $N$ is the total number of robots in swarm, that is, 15 to 30 robots; $P_r$ is the price of extended version of e-puck without range-&-bearing modules, that is, 2,000€; $P_x$ and $P_y$ are the prices of a range-&-bearing receiver and a range-&-bearing transmitter respectively—see Table 3.

The minimum cost of a swarm is 43,500€, when the minimum number of 15 robots are equipped with the least-capable range-&-bearing receiver and transmitter modules. The maximum cost of a swarm is 108,000€, when the maximum number of 30 robots are equipped with the most-capable range-&-bearing receiver and transmitter modules.

For each mission, we perform two experiments, $M_{80}$ and $M_{60}$, where the monetary budget is 80,000€ and 60,000€ respectively—see Table 4.

### Power-constraint

In these experiments, the limited resource is the battery capacity, $P_{bc}$. There is no constraint on the monetary resources: the design process can choose any combination of the range-&-bearing modules and the number of robots between 15 and 30—see Table 4. The operation time, $T_r$, of each robot in the swarm depends on its hardware configuration, available battery capacity, and the execution of the individual-level behaviors. The operation time

of a robot can be computed by the following equation:

$$T_r = \frac{(P_{bc} \times 3600)}{(I_{cpu} + I_{lm} + I_{rm} + I_{y(\mathcal{R})} + I_x)}, \quad (6)$$

where $I_{cpu}$ is the current rating of robot's cpu and other fixed sensors, that is, 100 mA. The CPU and other fixed hardware modules will always consume the same power. $I_{lm}$ and $I_{rm}$ are the current ratings of the left and the right motors of the robot, that is, 150 mA at maximum speed. $I_x$ and $I_{y(\mathcal{R})}$ are the current ratings of range-&-bearing receiver and transmitter modules respectively. $\mathcal{R}$ is the range of range-&-bearing transmitter—see Table 3. The experiment terminates, when every robot in the swarm consumes its battery power. The total experiment time, $T$, is expressed as:

$$T = \max\left(T_{r \in \{1,2,3,\dots,N\}}\right). \quad (7)$$

For each mission, we perform two experiments with different levels of battery capacities: $P_{20}$ and $P_{15}$—see Table 4.

### Monetary-&-Power-Constraint

In these experiments, both monetary budget and battery capacity are limited. The design process is required to choose the hardware modules that are not only affordable but also allow robots to operate for a sufficient amount of time. For each mission, we perform four experiments with dual constraints: $M_{80}P_{20}$, $M_{80}P_{15}$, $M_{60}P_{20}$, and $M_{60}P_{15}$—see Table 4.

### Protocol

The experiments are performed without any human intervention. The design of control software and the configuration of hardware is the result of an automatic design process. For each experiment, we run 20 independent design processes to get 20 hardware configurations and their respective control software in the form of a finite state machine. Each design process is run with the design budget of 50,000 simulations. The performance of the designs are evaluated via a single run of each design. For each experiment, we report (i) the performance achieved by the swarm, the number of robots comprised in the automatically designed swarm, the hardware modules that have been automatically selected, and the adopted software modules.

## RESULTS

In this section, we present the results on a per-mission basis. The instances of control software generated, the data collected, and videos of the experiments are available as online supplementary material (*Salman, Ligot & Birattari, 2019*).

### ANYTIME-SELECTION

The performance of an automatically designed swarm depends on the number of robots that reach the black zone and on the moment in which each of them does so: the longer a robot remains on the black zone, the higher the contribution it makes to the score. As a result, the duration of the experiment has an impact on the performance: the longer the experiment, the longer the robot can remain on the black zone and contribute to the score.

When economical constraints are imposed, the design process tends to select low-tier hardware; and designs the control software such that the robots move less and save battery life for a longer experiment duration.

### No-constraint

`Waffle` tends to configure robot swarms whose total cost is close to the maximum possible—see Fig. 3D. Indeed, the robot swarms comprise 25 to 30 robots—see Fig. 3G—equipped with high-tier range-&-bearing receivers and transmitters—see Fig. 4. At visual inspection, the robots first form clusters and then slowly converge on a black zone: when the robots find a black zone, they spin in place and block the way for the remaining robots, which are therefore unable to enter the zone. This behavior is obtained with Exploration, Stop, and Attraction—see Fig. 5A. As expected, the performance of the swarm is considerably better than the one achieved when constraints are imposed—see Fig. 3A.

### Monetary-constraint

Under the constraints imposed by $M_{80}$ and $M_{60}$, `Waffle` tends to configure the robot swarm so that the total cost is close to the maximum available budget—see Fig. 3D. The number of robots in the swarm decreases proportionally to the monetary budget—see Fig. 3G. The robots are equipped with high-tier range-&-bearing receivers and long-range range-&-bearing transmitters. In $M_{60}$, however, `Waffle` also selects two low-tier range-&-bearing receivers—see Fig. 4. The robot swarms designed under $NC$ and $M_{80}$ behave in a similar way. In $M_{60}$, however, the robots prefer to use the Attraction low-level behavior to remain in a black zone, but as the robots are equipped with low-tier range-&-bearing receivers, they often leave the black zone: due to the high loss-probability of low-tier range-&-bearing receivers, the robots often fail to perceive the presence of their peers in their neighborhood. The performance of the swarms designed under $M_{80}$ and $M_{60}$ is considerably lower than the one achieved under $NC$: in $M_{80}$ and $M_{60}$, the swarm comprises fewer robots as compared to $NC$—see Fig. 3A.

### Power-constraint

In contrast to $NC$, the swarms configured under $P_{20}$ and $P_{15}$ have a total cost that is noticeably lower than the maximum possible—see Fig. 3D. This is because the robots are equipped with low-price range-&-bearing transmitters, which reduces the overall cost of the swarm—see Fig. 3D. Cheaper range-&-bearing transmitters have a shorter transmission range but have low power consumption, which allows a longer battery life. We observe a major shift in the dominant individual-level behaviors in the produced instances of control software. The robots stop in the first black zone they encounter and limit their movement to save energy—see Fig. 5A. Consequently, the swarm splits and becomes unable to gather on the same zone. As a result, the performance drops by approximately 50% as compared to the performance achieved under $NC$—see Fig. 3A.

### Monetary-&-power-constraint

In all experiments, `Waffle` tends to use all the available monetary budget—see Fig. 3D. In $M_{80}P_{20}$ and $M_{80}P_{15}$, `Waffle` designs swarms that comprise 23 to 24 robots—see Fig. 3G—equipped with any of the range-&-bearing receivers and low-range transmitters. In $M_{60}P_{20}$

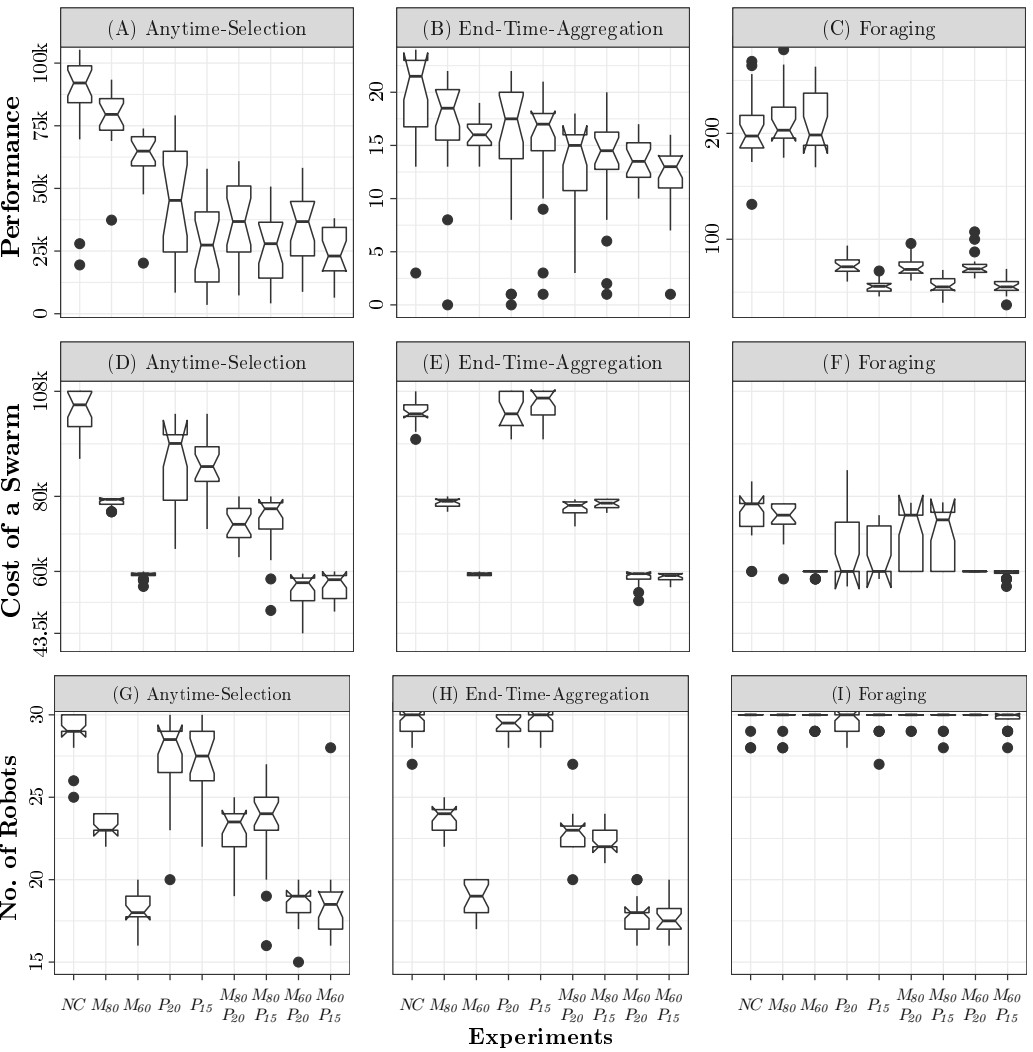

**Figure 3** **The performance in all nine experiments on each mission is shown at the top.** Performance of Waffle in all nine experiments for each mission: ANYTIME-SELECTION (A), END-TIME-AGGREGATION (B), and FORAGING (C). Total cost of the swarms configured by Waffle for each mission (D–F): 43.5k and 108k are the minimum and maximum possible cost (in €) of a swarm, respectively. Number of robots selected by Waffle for each mission (G–I).

and $M_{60}P_{15}$; however, the number of robots decrease considerably, and the robots are equipped with low-tier range-&-bearing receivers and low-range transmitters—see Figs. 3G and 4. The control software generated under the *Monetary-&-Power-Constraint* behave similarly to those of the *Power-Constraint* experiments—see Fig. 5A. Due to the increased battery capacity, the swarms produced under $M_{80}P_{20}$ and $M_{60}P_{20}$ perform slightly better than the ones produced under $P_{15}$, $M_{80}P_{15}$, and $M_{60}P_{15}$—see Fig. 3A.

### END-TIME-AGGREGATION

The performance of a designed swarm depends solely on the number of robots that are on the black zone at the end of an experiment. Contrary to Anytime-Selection, if economical

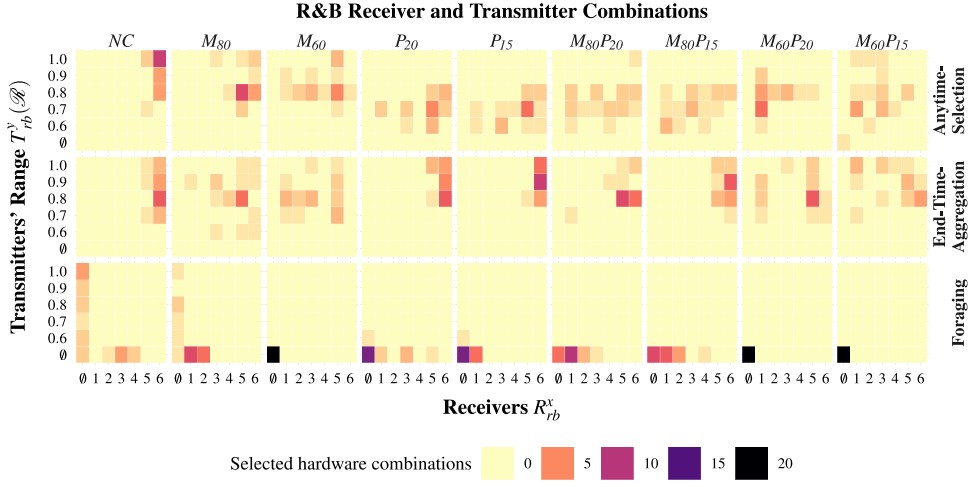

**Figure 4 The number of instances of a specific hardware combination selected in each experiment is shown here.** The horizontal axis represents the possible configurations of the range-&-bearing receivers $R_{rb}^x$; the vertical one represents the possible ranges $\mathcal{R}$ of the range-&-bearing transmitters $T_{rb}^y$. Here, Ø represents the case in which the design process does not select any range-&-bearing receiver or transmitter, $x \in \{\emptyset, 1, 2, 3, 4, 5, 6\}$, and $y \in \{\emptyset, 1, 2\}$ as shown in Table 3.

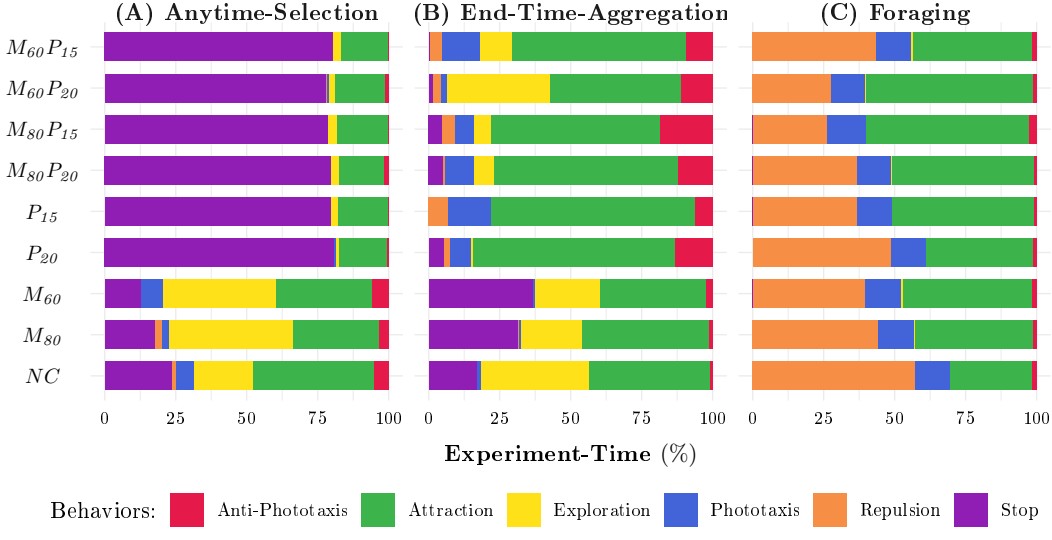

**Figure 5 Behaviors adopted by the robots in the experiments for the three missions considered: ANYTIME-SELECTION (A), END-TIME-AGGREGATION (B), and FORAGING (C).** Each color represents a behavior. The videos of all experiments are available as online supplementary material (*Salman, Ligot & Birattari, 2019*).

constraints are applied, the design process tends to select high-tier hardware; and the control software is composed of individual-level behaviors that keep robots assembled on a black zone.

### No-constraint

`Waffle` tends to configure robot swarms whose total cost is close to the maximum possible—see Fig. 3E. The hardware configuration is similar to the one generated under *NC* for Anytime-Selection. Indeed, the robot swarms comprise 28 to 30 robots—see Fig. 3H—equipped with high-tier range-&-bearing receivers and transmitters—see Fig. 4. At visual inspection, the robots first form clusters and then converge on a black zone: robots tend to remain there by spinning in place until the end of the experiment. This behavior is obtained with Exploration, Attraction, and Stop—see Fig. 5B. The performance of the swarm is considerably better than those achieved when constraints are imposed—see Fig. 3B.

### Monetary-constraint

Under the constraints imposed by $M_{80}$ and $M_{60}$, `Waffle` tends to configure the robot swarm so that the total cost is close to the maximum available budget—see Fig. 3E. The number of robots in the swarm decreases proportionally to the monetary budget—see Fig. 3H. The robots are equipped with long-range range-&-bearing transmitters and high-tier receivers, except a small minority of configurations in which the robots are equipped with low-tier receivers—see Fig. 4. At visual inspection, in $M_{80}$ and $M_{60}$ the robots converge on a black zone and stay there until the end of the experiments. Contrary to *NC*, the robots stop on the black zone instead of spinning in place: dominant individual-level behaviors are Exploration, Attraction, and Stop—see Fig. 5B. The amount of available monetary budget has a direct impact on the performance of a swarm. Indeed, due to the limited monetary budget, the number of robots decreases in the swarms designed under $M_{80}$ and $M_{60}$, which results in a considerable performances drop as compared to the performance achieved under *NC*—see Fig. 3B.

### Power-constraint

Similar to *NC*, under the constraints imposed by $P_{20}$ and $P_{15}$, `Waffle` tends to configure the robot swarm so that the total cost is close to the maximum possible—see Fig. 3E. Indeed, the robot swarms comprise 28 to 29 robots—see Fig. 3H—equipped with high-tier range-&-bearing receivers and long-range transmitters—see Fig. 4. However, this selection of hardware has a direct impact on the duration of the experiments due to its high current rating. As the maximum power is consumed by the motors, the designed control software skips the Exploration behavior to move robots in the arena. In some instances of control software, the robots use Phototaxis and Anti-Phototaxis individual-level behaviors to move straight and avoid obstacles. Moreover, the most dominant individual-level behavior is Attraction which is used to keep the robots assembled on one zone—see Fig. 5B. The performance achieved under $P_{20}$ is relatively higher than the one achieved under $P_{15}$. Due to the limited battery capacity, which affects the total experiment duration, the swarms designed under $P_{20}$ and $P_{15}$ have a lower performance than those designed under *NC*—see Fig. 3B.

*Monetary-&-power-constraint*

In all experiments, `Waffle` tends to use all the available monetary budget—see Fig. 3E. In $M_{80}P_{20}$ and $M_{80}P_{15}$, `Waffle` designs swarms that comprise 22 to 24 robots—see Fig. 3H—equipped with high-tier range-&-bearing receivers and long-range transmitters—see Fig. 4. In $M_{60}P_{20}$ and $M_{60}P_{15}$, the number of robots decreases considerably, and the robots are equipped with high-tier range-&-bearing receivers and long-range transmitters, except a small minority of configurations in which robots are equipped with low-tier receivers—see Fig. 4. The instances of control software produced are similar to those produced under *Power-Constraint*: the movement of robots in the arena is identical and the prominent individual-level behavior is Attraction—see Fig. 5B. The performance achieved under $M_{80}P_{20}$ and $M_{80}P_{15}$ is slightly better than the one achieved under $M_{60}P_{20}$ and $M_{60}P_{15}$: the level of monetary budget is the key factor that determines whether `Waffle` selects few or more robots—see Fig. 3B.

### FORAGING

Similar to ANYTIME-SELECTION, the performance of swarms designed in the FORAGING experiments depends on the experiment duration, but it also depends on the total number of robots. Contrary to both ANYTIME-SELECTION and END-TIME-AGGREGATION, the individual robots do not rely on the range-&-bearing hardware. The control software produced enables an effective movement between source and nest.

*All categories of constraints*

Under all the categories of constraints considered, `Waffle` produced robot swarms sharing the same hardware configuration. This because, in FORAGING, the robots do not rely on local communication. As a result, the selected hardware configuration typically does not include range-&-bearing transmitter and receiver—see Fig. 4. The total cost of a swarm is between 80,000€ and 60,000€—see Fig. 3F. The swarm comprises the largest possible number of robots—see Fig. 3I.

All instances of control software that are produced in all experiments have an unexpected behavior. Although in all experiments of FORAGING, the robots are not equipped with range-&-bearing modules, the most prominent individual-level behaviors are Attraction and Repulsion, which the robots use to explore the arena—see Fig. 5C. The swarm uses these behaviors in a way that is completely different from the one originally intended (*Francesca et al., 2014*). The reason behind this anomaly is that the individual-level behaviors in the design space are not strictly associated with the related hardware. In the absence of range-&-bearing receivers and transmitters, the Attraction and Repulsion behaviors are actuating robots to move straight using proximity sensors to avoid obstacles. In all FORAGING experiments, `Waffle` selects the Phototaxis individual-level behavior to locate the nest in the arena, as shown in Fig. 5C.

There is no prominent performance difference between the experiments under the *No-Constraint* and *Monetary-Constraint* categories—see Fig. 3C. However, we observe a considerable performance drop by the swarms designed under categories of experiments that have limited battery capacity—that is, *Power-Constraint* and *Monetary-&-Power-Constraint*. Indeed, the performance achieved in experiments with 20 mAh battery

capacity—i.e., $P_{20}$, $M_{80}P_{20}$, and $M_{60}P_{20}$—is considerably better than the performance achieved under experiments with 15 mAh battery capacity—i.e., $P_{15}$, $M_{80}P_{15}$, and $M_{60}P_{15}$—see Fig. 3C.

## CONCLUSIONS

In this paper, we studied the concurrent automatic design of control software and the automatic configuration of the hardware of robot swarms. In particular, we showed that it is possible to concurrently design control software and hardware for a robot swarm using the principles of automatic modular design. We introduced Waffle, a new instance of the AutoMoDe family of automatic design methods that configures the robot hardware, selects the number of robots in the swarm, and produces control software in the form of a probabilistic finite state machine by combining pre-existing modules that are mission independent. We studied Waffle under economic constraints on the total monetary budget available and on the battery capacity of each individual robot comprised in the swarm. We tested Waffle on three different collective missions. In the experiments presented in the paper, Waffle was able to concurrently design the control software and configure the hardware of a robot swarm. The results suggest that the hardware configuration of the individual robots, the design of control software, and the number of robots highly depend on the nature of the collective mission and the economical constraints imposed. In the paper, we only considered the automatic configuration of one type of hardware module, future studies will focus on extending the automatic design to other sensors and actuators. The range-&-bearing receivers and transmitters proposed in the paper can be manufactured and real-robot experiments can be performed to assess the robustness of the selected configuration to the reality gap.

### Funding

The research has received funding from the European Research Council (ERC) under the European Union's Horizon 2020 research and innovation programme (grant agreement No 681872). Mauro Birattari received support from the Belgian Fonds de la Recherche Scientifique – FNRS. The funders had no role in study design, data collection and analysis, decision to publish, or preparation of the manuscript.

### Grant Disclosures

The following grant information was disclosed by the authors:
European Research Council (ERC): 681872.
Belgian Fonds de la Recherche Scientifique – FNRS.

### Competing Interests

Mauro Birattari is an Academic Editor for PeerJ.

## Author Contributions

- Muhammad Salman conceived and designed the experiments, performed the experiments, analyzed the data, prepared figures and/or tables, performed the computation work, authored or reviewed drafts of the paper, approved the final draft.
- Antoine Ligot performed the computation work, authored or reviewed drafts of the paper, approved the final draft.
- Mauro Birattari contributed reagents/materials/analysis tools, conceived and designed the experiments, analyzed the data, directed the research, authored or reviewed drafts of the paper, approved the final draft.

## Data Availability

The data is available at IRIDIA - Supplementary Information, ID: IridiaSupp2019-001: http://iridia.ulb.ac.be/supp/IridiaSupp2019-001/.

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
