# Peer review of "Concurrent design of control software and configuration of hardware for robot swarms under economic constraints"

_PeerJ Computer Science, doi:10.7717/peerj-cs.221_

## Round 0.1 · original submission · Minor Revisions

I have much pleasure in letting you know that your paper has been now reviewed and both reviewers consider that your work is an interesting contribution for the journal provided minor modifications of the manuscript will be made. Please take into consideration the remarks of both reviewers. In my opinion all the comments have been very constructive, and they are an opportunity to improve the paper. Please try to refine the main contribution as stated by reviewer #1 and improve Table 2 and the additional figure suggested by reviewer #2.

I hope to receive soon a modified manuscript addressing the suggestions of the reviewers.

Reviewer 1 ·

Basic reporting

Well written, well structured. The authors are professionals and this shows. Background and context are OK, but I noted one missing reference that evolves sensory layout on a swarm of real robots.

https://www.researchgate.net/publication/290169158_Evolution_Individual_Learning_and_Social_Learning_in_a_Swarm_of_Real_Robots

Figures and Tables are well made.

The main point of criticism is about the contributions stated in the Intro. That offers two contributions, of which the second one “(ii) we generalize the notion of automatic design of robot swarms to include economic constraints.” is heavily overclaimed. What I mean is the ALL design problems are subject to multiple constraints. Furthermore, constraints can be divided into several categories. Including economic constraints is NOT generalizing the notion of (automatic) design. It merely considers constraints (one, to be precise) from the category Economic.

Experimental design

The main RQ is "Can Waffle select mission-specific hardware together with an appropriate control software?"
and this is a relevant one. Note that the question concerns a particular SYSTEM (Waffle), not a generic algorithm or methodology.

The experiments are well designed and conducted. I have one question, though, Table 4 shows specific values of parameters - how were these values determined?

Validity of the findings

The main RQ was "Can Waffle select mission-specific hardware together with an appropriate control software?"
and indeed
"In the experiments presented in the paper, Waffle was able to concurrently design the control software and configure the hardware of a robot swarm."

This leads to revisiting the first contribution: "we address the concurrent automatic design of control software and the automatic configuration of the hardware". I would prefer to sharpen this to what the paper really does and demonstrates.

Additional comments

Good work, good paper, but the value is blurred by the somewhat misplaced / misphrased main contributions.

·

Basic reporting

The paper has a good presentation quality with sufficient background.

Experimental design

The paper's relevance lies in the novel automatic "configuration" of hardware. The use of extensive simulation suffice.

Validity of the findings

Results are well presented. Particularly, Figure 5 is very useful and provides an interesting overview of the automatically obtained controller.

Additional comments

The work extends AutoMoDe to include the automatic "configuration" of hardware alongside the existing automatic design of control software. The proposed hardware configuration involves the selection of appropriate range-&-bearing receivers and transmitters as well as the selection of the number of robots in the swarm subject to battery and economic constraints. Although the work is carried under simulation and using hypothetical range-&-bearing modules, it represents a valuable insight into the automatic selection of hardware modules for swarm robotics and it is worthy of publication.

I provide below a few minor suggestions(+) and strongly encouraged changes (*), which I hope will improve the paper:

+ I am not sure that hardware “configuration” is the best term, as it seems that the automatic design select appropriate hardware modules/components.

+ In the introduction it could be beneficial to include an explanation of what are constraints and what are objective functions in an optimisation context. Then later, when you talk about economic and battery constraints mention that the objective function depends on the task that will be solved.

(*) Page 5 lines 176-177: The authors state that “In a probabilistic finite state machine, states are instances of behaviours and edges are instances of conditions” and this is not true for all probabilistic FSM. See the difference between Mealy and Moore machines, and check the following work for another perspective (citation to the work not required):

Lopes et al. Probabilistic supervisory control theory (psct) applied to swarm robotics. AAMAS 2017.

Thus, the authors may want to state that in their case states are associated with behaviours and transitions are associated with conditions, but not state that this is always the case.

+ Page 5 lines 203-204 (Table 3) states 6 receivers and 2 transmitters, but notice that $T^1$ has 3 range options and $T^2$ has 2 range options. When looking in Figure 4 this gets non-intuitive as the x-axis relates to the 6 receivers plus no receiver and the y-axis states the range. Maybe present it as 6 transmitters would be more clear.

+ Figure 1 is a quite unconventional way to represent a finite state machine (automaton). Usually, double-line circles represent marked states and an initial state should be indicated by an unlabelled arrow. The transitions could be represented by labelled arrows without the lozenge. I would also suggest describing the symbols in the figure's caption.

(*) Table 2. Please provide a detailed description of the components of the aggregate position (V).

(*) I also would like to see 2 or 3 scenarios (shown in a figure) and the resulting aggregate position (V), as defined in Equation 1, this would help to understand the meaning of this reading.

+ Page 4, line 182 it would be nice to include the version of the e-puck under consideration. Note that since 2018 e-puck2 is available, but it seems that version 1 is used.

+ Regarding the objective functions in Equations 2 and 3, I am curious about what would be the implication of divide the summation by the number of robots, so we have the ration of robots instead of the absolute quantity. How is the impact on the number of robots & hardware selected? I risk speculating that if the control software is poor it would tend to select a lower (and perhaps odd) number of robots, as it tends to obtain an average 50%-50% distribution between the black areas (7-8)/15 = 1/15 which would be bigger than (14-15)/29 = 1/29. However, a controller able to aggregate most of the robots in a particular black area as long as it has a “good” (and therefore more expensive) hardware may also select a low number of robots to attempt obtaining 15/15 = 1.0. This would be different from the current results that tend to select many robots. In principle simulations under this new objective function would be ideal, however, this could be discussed and presented as future work.

+ In table 4 you may consider using “Unconstrained” instead of “unlimited”

+ Page 9, line 309: add “on” --> “...swarm depends not only on the formation ...”

---

## Round 0.2 · accepted · Accept

The issues raised by the reviewers have been successfully addressed and I am pleased to accept the manuscript